# FORK: A FORward-looKing Actor for Model-Free Reinforcement Learning

## Abstract

In this paper, we propose a new type of Actor, named forward-looking Actor or FORK for short, for Actor-Critic algorithms. FORK can be easily integrated into a model-free Actor-Critic algorithm. Our experiments on six Box2D and MuJoCo environments with continuous state and action spaces demonstrate significant performance improvement FORK can bring to the state-of-the-art algorithms. A variation of FORK can further solve Bipedal-WalkerHardcore in as few as four hours using a single GPU.

## 1 Introduction

Deep reinforcement learning has had tremendous successes, and sometimes even superhuman performance, in a wide range of applications including board games (Silver et al., 2016), video games (Vinyals et al., 2019), and robotics (Haarnoja et al., 2018a). A key to these recent successes is the use of deep neural networks as high-capacity function approximators that can harvest a large amount of data samples to approximate high-dimensional state or action value functions, which tackles one of the most challenging issues in reinforcement learning problems with very large state and action spaces.

Many modern reinforcement learning algorithms are model-free, so they are applicable in different environments and can readily react to new and unseen states. This paper considers model-free reinforcement learning for problems with continuous state and action spaces, in particular, the Actor-Critic method, where Critic evaluates the state or action values of the Actor's policy and Actor improves the policy based on the value estimation from Critic. To draw an analogy between Actor-Critic algorithms and human decision making, consider the scenario where a high school student is deciding on which college to attend after graduation. The student, like Actor, is likely to make her/his decision based on the perceived values of the colleges, where the value of a college is based on many factors including (i) the quality of education it offers, its culture, and diversity, which can be viewed as instantaneous rewards of attending the college; and (ii) the career opportunities after finishing the college, which can be thought as the future cumulative reward. We now take this analogy one step further, in human decision making, we often not only consider the "value" of *current* state and action, but also further forecast the outcome of the current decision and the value of the *next* state. In the example above, a student often explicitly takes into consideration the first job she/he may have after finishing college, and the "value" of the first job. Since forward-looking is common in human decision making, we are interested in understanding whether such forward-looking decision making can help Actor; in particular, whether it is useful for Actor to forecast the next state and use the value of future states to improve the policy. To our great surprise, a relative straightforward implementation of forward-looking Actor, as an add-on to existing Actor algorithms, improves Actor's performance by a large margin.

Our new Actor, named FOrward-looKing Actor or FORK for short, mimics human decision making where we think multi-step ahead. In particular, FORK includes a neural network that forecasts the next state given the current state and current action, called *system network*; and a neural network that forecasts the reward given a (state, action) pair, called *reward network*. With the system network and reward network,

FORK can forecast the next state and consider the value of the next state when improving the policy. For example, consider the Deep Deterministic Policy Gradient (DDPG) (Lillicrap et al., 2015), which updates the parameters of Actor as follows:

$$\phi \leftarrow \phi + \beta \nabla_\phi Q_\psi(s_t, A_\phi(s_t)),$$

where $s_t$ is the state at time $t$, $\phi$ are Actor's parameters, $\beta$ is the learning rate, $Q_\psi(s, a)$ is the Critic network, and $A_\phi(s)$ is the Actor network. With DDPG-FORK, the parameters can be updated as follows:

$$\phi \leftarrow \phi + \beta \left( \nabla_\phi Q_\psi(s_t, A_\phi(s_t)) + \nabla_\phi R_\eta(s_t, A_\phi(s_t)) + \gamma \nabla_\phi R_\eta(\tilde{s}_{t+1}, A_\phi(\tilde{s}_{t+1})) + \right.$$
$$\left. \gamma^2 \nabla_\phi Q_\psi(\tilde{s}_{t+2}, A_\phi(\tilde{s}_{t+2})) \right), \tag{1}$$

where $R_\eta$ is the reward network, and $\tilde{s}_{t+1}$ and $\tilde{s}_{t+2}$ are the future states forecast by the system network $F_\theta$.

We will see that FORK can be easily incorporated into most deep Actor-Critic algorithms, by adding two additional neural networks (the system network and the reward network), and by adding extra terms to the loss function when training Actor, e.g. adding term

$$R_\eta(s_t, A_\phi(s_t)) + \gamma R_\eta(\tilde{s}_{t+1}, A_\phi(\tilde{s}_{t+1})) + \gamma^2 Q_\psi(\tilde{s}_{t+2}, A_\phi(\tilde{s}_{t+2}))$$

for each sampled state $s_t$ to implement (1).

We remark that Equation (1) is just one example of FORK, FORK can have different implementations (a detailed discussion can be found in Section 3). We further remark that learning the system model is not a new idea and has a long history in reinforcement learning, called model-based reinforcement learning (some state-of-the-art model-based reinforcement learning algorithms and the benchmark can be found in (Wang et al., 2019)). Model-based reinforcement learning uses the model in a sophisticated way, often based on deterministic or stochastic optimal control theory to optimize the policy based on the model. FORK only uses the system network as a blackbox to forecast future states, and does not use it as a mathematical model for optimizing control actions. With this key distinction, any model-free Actor-Critic algorithm with FORK remains to be model-free.

In our experiments, we added FORK to two state-of-the-art model-free algorithms, according to recent benchmark studies (Duan et al., 2016a; Wang et al., 2019): TD3 (Fujimoto et al., 2018) (for deterministic policies) and SAC (Haarnoja et al., 2018b) (for stochastic policies). The evaluations on six challenging environments with continuous state space and action space show significant improvement when adding FORK. In particular, TD3-FORK performs the best among the all we tested. For Ant-v3, it improves the average cumulative reward by more than $50\%$ than TD3, and achieves TD3's best performance using only $35\%$ of training samples. BipedalWalker-v3 is considered "solved" when the agent obtains an average cumulative reward of at least 300[1]. TD3-FORK only needs 0.23 million actor training steps to solve the problem, half of that under TD3. Furthermore, a variation of TD3-FORK solves BipedalWalkerHardcore, a well known difficult environment, with as few as four hours using a single GPU.

## 1.1 RELATED WORK

The idea of using learned models in reinforcement learning is not new, and actually has a long history in reinforcement learning. At a high level, FORK shares a similar spirit as model-based reinforcement learning and rollout. However, in terms of implementation, FORK is very different and much simpler. Rollout in general requires the Monte-Carlo method (Silver et al., 2017) to simulate a finite number of future states from the current state and then combines that with value function approximations to decide the action to take at the current time. FORK does not require any high-fidelity simulation. The key distinction between FORK and model-based reinforcement learning is that model-based reinforcement learning uses the learned

---

[1]https://github.com/openai/gym/blob/master/gym/envs/box2d/bipedal_walker.py

model in a sophisticated manner. For example, in SVG (Heess et al., 2015), the learned system model is integrated as a part of the calculation of the value gradient, in (Gu et al., 2016), refitted local linear model and rollout are used to derive linear-Gaussian controller, and (Bansal et al., 2017) uses a learned dynamical model to compute the trajectory distribution of a given policy and consequently estimates the corresponding cost using a Bayesian optimization-based policy search. More model-based reinforcement learning algorithms and related benchmarking can be found in (Wang et al., 2019). FORK, on the other hand, only uses the system network to predict future states, and does not use the system model beyond that. Other related work that accelerates reinforcement learning algorithms includes: acceleration through exploration strategies (Gupta et al., 2018), optimizers (Duan et al., 2016b), and intrinsic reward (Zheng et al., 2018), just to name a few. These approaches are complementary to ours. FORK can be added to further accelerate learning.

## 2 BACKGROUND

Reinforcement Learning algorithms aim at learning policies that maximize the cumulative reward by interacting with the environment. We consider a standard reinforcement learning setting defined by a Markov decision process (MDP) $(\mathcal{S}, \mathcal{A}, p_0, p, r, \gamma)$, where $\mathcal{S}$ is a set of states, $\mathcal{A}$ is the action space, $p_0(s)$ is the probability distribution of the initial state, $p : \mathcal{S} \times \mathcal{S} \times \mathcal{A} \to [0, \infty)$ is the transition density function, which represents the distribution of the next state $s_{t+1}$ given current state $s_t$ and action $a_t$, $r : \mathcal{S} \times \mathcal{A} \to [r_{\min}, r_{\max}]$ is the bounded reward function on each transition, and $\gamma \in (0, 1]$ is the discount factor. We consider a discrete-time system. At each time step $t$, given the current $s_t \in \mathcal{S}$, the agent selects an action $a_t \in \mathcal{A}$ based on a (deterministic or stochastic) policy $\pi(a_t|s_t)$, which moves the environment to the next state $s_{t+1}$, and yields a reward $r_t = r(s_t, a_t)$ to the agent. We consider stationary policies in this paper under which the action is taken based on $s_t$, and is independent of other historical information.

Starting from time 0, the return of given policy $\pi$ is the discounted cumulative reward

$$J_\pi(i) = \sum_{t=0}^{T} \gamma^t r(s_t, a_t), \quad \text{given } s_0 = i.$$

$J_\pi(i)$ is also called the state-value function. Our goal is to learn a policy $\pi^*$ that maximizes this cumulative reward

$$\pi^* \in \arg\max_\pi J_\pi(i) \quad \forall i.$$

We assume our policy is parameterized by parameter $\phi$, denoted by $\pi_\phi$, e.g. by the Actor network in Actor-Critic Algorithms. In this case, our goal is to identify the optimal parameter $\phi^*$ that maximizes

$$\phi^* \in \arg\max J_{\pi_\phi}(i).$$

Instead of state-value function, it is often convenient to work with action-value function, Q-function, which is defined as follows:

$$Q_\pi(s, a) = E\left[r_\pi(s, a) + \gamma J_\pi(s')\right],$$

where $s'$ is the next state given current state $s$ and action $a$. The optimal policy is a policy that satisfies the following Bellman equation (Bellman, 1957):

$$Q_{\pi^*}(s, a) = E\left[r(s, a) + \gamma \max_{a' \in \mathcal{A}} Q_{\pi^*}(s', a')\right].$$

When neural networks are used to approximate action-value functions, we denote the action-value function by $Q_\psi(s, a)$, where $\psi$ is the parameters of the neural network.

## 3 FORK — FORWARD-LOOKING ACTOR

This paper focuses on Actor-Critic algorithms, where Critic estimates the state or action value functions of the current policy, and Actor improves the policy based on the value functions. We propose a new type of Actor, FORK. More precisely, a new training algorithm that improves the policy by considering not only the action-value of the current state (or states of the current mini-batch), but also future states and actions forecast using a learned system model and a learned reward model. This forward-looking Actor is illustrated in Figure 1. In FORK, we introduce two additional neural networks:

**The system network $F_\theta$.** The network is used to predict the next state of the environment, i.e., given current state $s_t$ and action $a_t$, it predicts the next state $\tilde{s}_{t+1} = F_\theta(s_t, a_t)$. With experiences $(s_t, a_t, s_{t+1})$, training the system network is a supervised learning problem. The neural network can be trained using mini-batch from replay-buffer and smooth-L1 loss $L(\theta) = \|s_{t+1} - F_\theta(s_t, a_t)\|_{\text{smooth L1}}$.

**The reward network $R_\eta$.** This network predicts the reward given current state $s_t$ and action $a_t$, i.e. $\tilde{r}_t = R_\eta(s_t, a_t)$. The network can be trained from experience $(s_t, a_t, r_t)$, with MSE loss $L(\eta) = \|r_t - R_\eta(s_t, a_t)\|^2$.

> Forward-Looking Actor
> $A_\phi(s)$   $F_\theta(s,a)$   $R_\eta(s, a)$
>
> Loss functions:
>
> Actor: $L(\phi) = E[-Q_\psi(s_t, A_\phi(s_t)) - R_\eta(s_t, A_\phi(s_t)) - \gamma R_\eta(s_{t+1}, A_\phi(\tilde{s}_{t+1})) - \gamma^2 Q_\psi(\tilde{s}_{t+2}, A_\phi(\tilde{s}_{t+2}))]$
>
> system network: $L(\theta) = E[\|s_{t+1} - F_\theta(s_t, a_t)\|_{\text{smooth-L1}}]$
>
> reward network: $L(\eta) = E[\|r_t - R_\eta(s_t, a_t)\|^2]$

Figure 1: FORK includes three neural networks, the policy network $A_\phi$, the system model $F_\theta$, and the reward model $R_\eta$.

**FORK.** With the system network and the reward network, the agent can forecast the next state, the next next stat and so on. Actor can then use the forecast to improve the policy. For example, we consider the following loss function

$$L(\phi) = E\left[-Q_\psi(s_t, A_\phi(s_t)) - R_\eta(s_t, A_\phi(s_t)) - \gamma R_\eta(\tilde{s}_{t+1}, A_\phi(\tilde{s}_{t+1})) - \gamma^2 Q_\psi(\tilde{s}_{t+2}, A_\phi(\tilde{s}_{t+2}))\right]. \quad (2)$$

In the loss function above, $s_t$ are from data samples (e.g. replay buffer), $\tilde{s}_{t+1}$ and $\tilde{s}_{t+2}$ are calculated from the system network as shown below:

$$\tilde{s}_{t+1} = F_\theta(s_t, A_\phi(s_t)) \quad \text{and} \quad \tilde{s}_{t+2} = F_\theta(\tilde{s}_{t+1}, A_\phi(\tilde{s}_{t+1})). \quad (3)$$

Note that when training Actor $A_\phi$ with loss function $L(\phi)$, all other parameters in $L(\phi)$ are regarded as constants except $\phi$ (see the PyTorch code in the supplemental materials).

The action-function $Q$, without function approximation, under current policy $A_\phi$ satisfies

$$Q(s_t, A_\phi(s)) = E\left[r(s_t, A_\phi(s_t)) + \gamma r(s_{t+1}, A_\phi(s_{t+1})) + \gamma^2 Q(s_{t+2}, A_\phi(s_{t+2}))\right],$$

where $r$, $s_{t+1}$ and $s_{t+2}$ are the actual rewards and states under the current policy, not estimated values. Therefore, the loss function $L(\phi)$ can be viewed as the average of two estimators.

Given action values from Critic and with a mini-batch of size $N$, FORK updates its parameters as follows:

$$\phi \leftarrow \phi - \beta_t \nabla_\theta L(\phi),$$

where $\beta_t$ is the learning rate and

$$\nabla_\phi L(\phi) = \frac{1}{N} \sum_{i=1}^{N} \left(\nabla_a Q_\psi(s_i, a)|_{a=A_\phi(s_i)} \nabla_\phi A_\phi(s_i) + \nabla_a R_\eta(s_i, a)|_{a=A_\phi(s_i)} \nabla_\phi A_\phi(s_i)\right.$$
$$\left. + \gamma \nabla_a R_\eta(\tilde{s}'_i, a)|_{a=A_\phi(\tilde{s}'_i)} \nabla_\phi A_\phi(\tilde{s}'_i). + \gamma^2 \nabla_a Q_\psi(\tilde{s}''_i, a)|_{a=A_\phi(\tilde{s}''_i)} \nabla_\phi A_\phi(\tilde{s}''_i)\right),$$

where $\tilde{s}'_i$ and $\tilde{s}''_i$ are the next state and the next next state estimated from the system network.

We note that it is important to use the system network to generate future states as in Equation (3) because they mimic the states under the current policy. If we would sample a sequence of consecutive states from the replay buffer, then the sequence is from the old policy, which does not help the learning. Figure 2 compares TD3-FORK, TD3, and TD3-MT, which samples a sequence of three consecutive states, on the BipedalWalker-v3 environment. We can clearly see that simply using consecutive states from experiences does not help improve learning. In fact, it significantly hurts the learning.

**Modified Reward Model:** We found from our experiments that the reward network can more accurately predict reward $r_t$ when including the next state $s_{t+1}$ as input into the reward network (an example can found in Appendix A.1). Therefore, we use a modified reward network $R_\eta(s_t, a_t, s_{t+1})$ in FORK.

**Adaptive Weight:** Loss function $L(\phi)$ in our algorithm uses the system network and the reward network to boost learning. In our experiments, we found that the forecasting can significantly improve the performance, except at the end of learning. Since the system and reward networks are not perfect, the errors in prediction can introduce errors/noises. To overcome this issue, we found it is helpful to use an adaptive weight $w$ so that FORK accelerates learning at the beginning but its weight decreases gradually as it gets close to the learning goal. A comparison between fixed weights and adaptive weights can

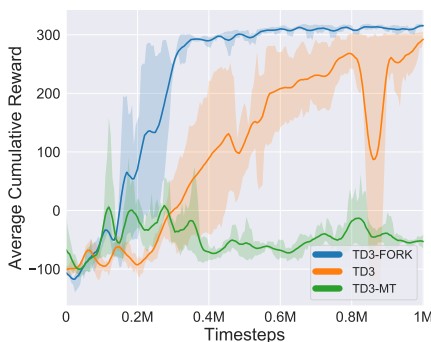

Figure 2: TD3-FORK versus TD3-MT

be found in Appendix A.2. We use a simple adaptive weight $w = \left(\frac{\bar{r}}{r_0}\right)_0^1 w_0$, where $\bar{r}$ is the moving average of cumulative reward (per episode), and $r_0$ is a predefined goal, $w_0$ is the initial weight, and $(a)_0^1 = a$ if $0 \le a \le 1$, $= 0$ if $a < 0$ and $= 1$ if $a > 1$. The loss function with adaptive weight becomes

$$L(\phi) = E\left[-Q_\psi(s_t, A_\phi(s_t)) - wR_\eta(s_t, A_\phi(s_t)) - w\gamma R_\eta(\tilde{s}_{t+1}, A_\phi(\tilde{s}_{t+1})) - w\gamma^2 Q_\psi\left(\tilde{s}_{t+2}, A_\phi(\tilde{s}_{t+2})\right)\right].$$
(4)

Furthermore, we set a threshold and let $w = 0$ if the loss of the system network is larger than the threshold. This is to avoid using FORK when the system and reward networks are very noisy. We note that in our experiments, the thresholds were chosen such that $w = 0$ for around $20,000$ steps at the beginning of each instance, which includes the first 10,000 random exploration steps.

**Different Implementations of FORK:** It is easy to see FORK can be implemented in different forms. For example, instead of two-step ahead, we can use one-step ahead as follows:

$$L(\phi) = E\left[-Q_\psi(s_t, A_\phi(s_t)) - wR_\eta(s_t, A_\phi(s_t)) - w\gamma Q_\psi\left(\tilde{s}_{t+1}, A_\phi(\tilde{s}_{t+1})\right)\right],$$
(5)

or only use future action values:

$$L(\phi) = E\left[-Q_\psi(s_t, A_\phi(s_t)) - w\left(Q_\psi\left(\tilde{s}_{t+1}, A_\phi(\tilde{s}_{t+1})\right) + w'Q_\psi\left(\tilde{s}_{t+2}, A_\phi(\tilde{s}_{t+2})\right)\right)\right].$$
(6)

We compared these two versions with FORK. The performance comparison can be found in Appendix B.3.

## 4  EXPERIMENTS

In this section, we evaluate FORK as an add-on to existing algorithms. We name an algorithm with FORK as algorithm-FORK, e.g. TD3-FORK or SAC-FORK. As an example, a detailed description of TD3-FORK can be found in Appendix A.3. We focused on two algorithms: TD3 (Fujimoto et al., 2018) and SAC (Haarnoja

et al., 2018b) because they were found to have the best performance among model-free reinforcement learning algorithms in recent benchmarking studies (Duan et al., 2016a; Wang et al., 2019). We compared the performance of TD3-FORK and SAC-FORK with TD3, SAC and DDPG (Lillicrap et al., 2015).

## 4.1 BOX2D AND MUJOCO ENVIRONMENTS

We selected six environments: BipedalWalker-v3 from Box2D (Catto, 2011), Ant-v3, Hooper-v3, HalfCheetah-v3, Humanoid-v3 and Walker2d-v3 from MuJoCo (Todorov et al., 2012) as shown in Figure 3. All these environments have continuous state spaces and action spaces.

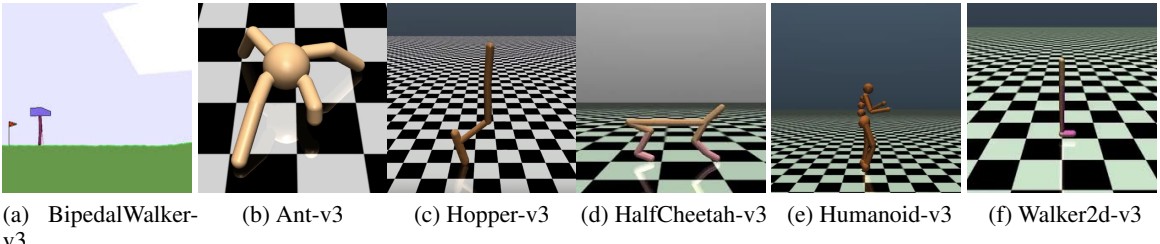

(a) BipedalWalker-v3    (b) Ant-v3    (c) Hopper-v3    (d) HalfCheetah-v3    (e) Humanoid-v3    (f) Walker2d-v3

Figure 3: The six environment used in our experiments

## 4.2 IMPLEMENTATION DETAILS

**Terminology. step (or time)**: one operation, e.g. training Actor with a mini-batch; **episode:** a single-run of the environment from the beginning to the end, consisting of many steps; and **instance:** the entire training consisting of multiple episodes.

**Hyperparameters.** Because FORK is an add-on, for TD3, we used the authors' implementation (`https://github.com/sfujim/TD3`); for SAC, we used a PyTorch version (`https://github.com/vitchyr/rlkit`) recommended by the authors *without any change except adding FORK*. The hyperparameters of both TD3 and SAC are summarized in Table 3 in Appendix A.4, and the hyperparameters related to FORK are summarized in Table 4 in the same appendix.

We can see TD3-FORK does not require much hyperparameter tuning. The system network and reward network used in the environments are the same except for the Humanoid-v3 for which we use larger system and reward networks because the dimension of the system is higher than other systems. The base weight $w_0$ is the same for all environments, the base rewards are the typical cumulative rewards under TD3 after a successful training, and the system thresholds are the typical estimation errors after about 20,000 steps.

SAC-FORK requires slightly more hyperparameter tuning. The base weights were chosen to be smaller values, the base rewards are the typical cumulative rewards under SAC, and the system thresholds are the same as those under TD3-FORK.

**Initial Exploration.** For each task and each algorithm, we use a random policy for exploration for the first 10,000 steps. Each step is one interaction with the environment.

**Duration of Experiments.** For each environment and each algorithm, we ran five different instances with different random seeds. Since we focus on Actor performance, Actor was trained for 0.5 million times for each instance. Since TD3 uses a delayed Actor with frequency 2 (i.e. Actor and Critic are trained with 1:2 ratio), Critic was trained one million times under TD3 and TD3-FORK. For SAC, SAC-FORK and DDPG,

Critic was trained 0.5 million times. The performance with the same amount of total training, including Critic training and Actor training, can be found in Appendix B.2, where for each algorithm, Critic and Actor, together, were trained 1.5 millions times.

## 4.3 RESULTS

Figure 4 shows the average cumulative rewards, where we evaluated the policies every 5,000 steps without exploration noise during training process. Each evaluation was averaged over 10 episodes. We train five different instances for each algorithm with same random seeds. The solid curves shows the average cumulative rewards (per episode), and the shaded region represents the standard deviations.

The best average cumulative rewards (its definition can be found in Appendix B.1) are summarized in Table 1. We can see that TD3-FORK outperforms all other algorithms. For Ant-v3, TD3-FORK improves the best average cumulative reward by more than 50% (5699.37 (TD3-FORK) versus 3652.11 (TD3)).

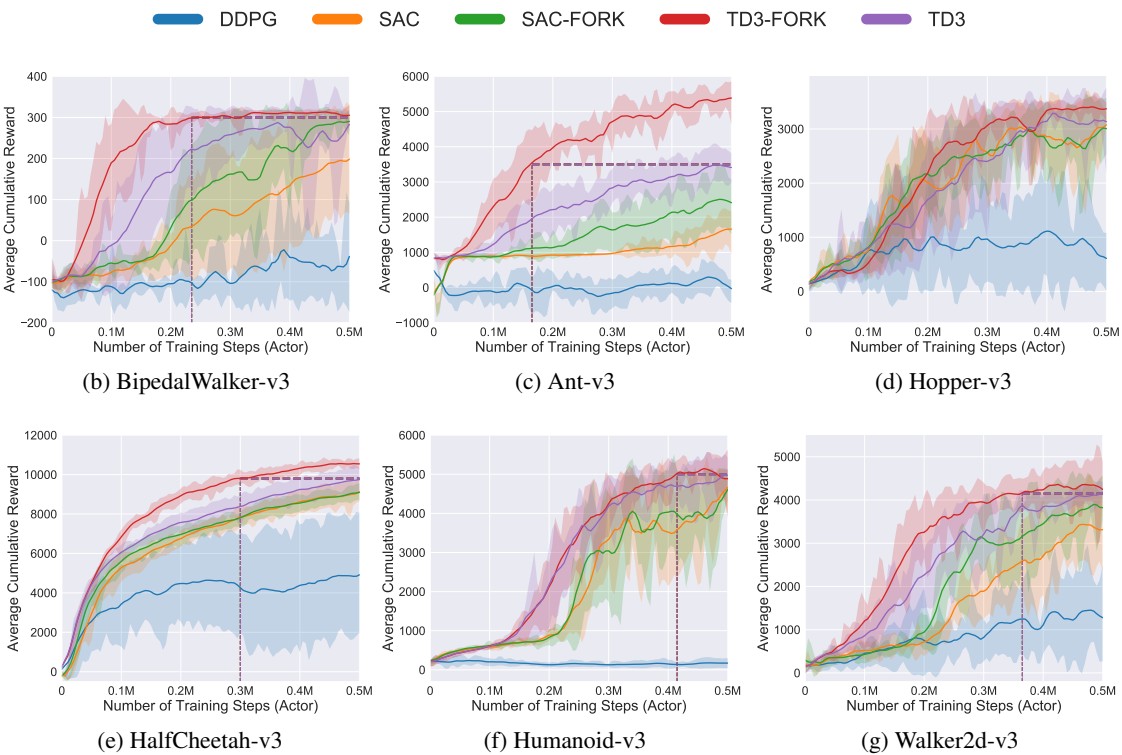

Figure 4: Learning curves of six environments. Curves were smoothed uniformly for visual clarity.

We also studied the improvement in terms of sample complexity. In Table 2, we summarized the number of Actor training required under TD3-FORK (SAC-FORK) to achieve the best average cumulative reward under TD3 (SAC). For example, for BipedalWalker-v3, TD3 achieved the best average cumulative reward with 0.4925 million steps of Actor training; and TD3-FORK achieved the same value with only 0.225 million steps of Actor training, reducing the required samples by more than a half.

Table 1: The best average cumulative rewards of the algorithms. The best value for each environment is highlighted in bold text.

| Environment | TD3-FORK | TD3 | DDPG | SAC | SAC-FORK |
|---|---|---|---|---|---|
| BipedalWalker-v3 | **317.40** | 307.69 | 125.90 | 223.20 | 303.18 |
| Ant-v3 | **5699.37** | 3652.11 | 995.59 | 1892.50 | 3091.00 |
| Hopper-v3 | **3522.57** | 3517.57 | 1713.52 | 3381.01 | 3359.18 |
| HalfCheetah-v3 | **10818.89** | 9893.04 | 5886.75 | 9314.64 | 8781.06 |
| Humanoid-v3 | **5439.31** | 5386.21 | 456.35 | 5293.89 | 5233.89 |
| Walker2d-v3 | **4616.46** | 4278.90 | 2404.60 | 4037.87 | 4243.69 |

Table 2: Sample complexity (million). The number of training steps needed for TD3-FORK (SAC-FORK) to achieve the same best average cumulative reward under TD3 (SAC). The numbers under TD3 (SAC) are the time steps at which the TD3 (SAC) achieved the best average cumulative rewards.

| Environment | TD3-FORK | TD3 | SAC-FORK | SAC |
|---|---|---|---|---|
| BipedalWalker-v3 | 0.225 | 0.492 | 0.370 | 0.495 |
| Ant-v3 | 0.272 | 0.475 | 0.285 | 0.495 |
| Hopper-v3 | 0.405 | 0.457 | 0.500+ | 0.335 |
| HalfCheetah-v3 | 0.295 | 0.495 | 0.500+ | 0.480 |
| Humanoid-v3 | 0.430 | 0.497 | 0.500+ | 0.495 |
| Walker2d-v3 | 0.360 | 0.500 | 0.396 | 0.475 |

In summary, FORK improves the performance of both TD3 and SAC after being included as an add-on. The improvement is more significant when adding to TD3 than adding to SAC. FORK improves TD3 in all six environments, and improves SAC in three of the six environments. Furthermore, TD3-FORK performs the best in all six environment. More statistics about this set of experiments can be found in Appendix B.1.

In Appendix B.2, we also presented experimental results where Actor and Critic together have the same amount of training across all algorithms (i.e. under TD3 and TD3-FORK, Actor was trained 0.5 million times and Critic was trained 1 million times; and under other algorithms, both Actor and Critic were trained 0.75 million times). In this case, TD3-FORK performs the best among four of the six environments, and SAC-FORK performs the best in the rest two environments.

### 4.4 BIPEDALWALKER-HARDCORE-V3

A variation of TD3-FORK can also solve a well-known difficult environment, BipedalWalker-Hardcore-v3, in as few as four hours using a single GPU. From the best of our knowledge, the known algorithm needs to train for days on a 72 cpu AWS EC2 with 64 worker processes taking the raw frames as input (`https://github.com/dgriff777/a3c_continuous`). You can view the performance on BipedalWalkerHardcore-v3 during and after training at `https://youtu.be/0nYQpXtxh-Q`. The implementation details can be found in Appendix C.

## 5 CONCLUSIONS

This paper proposes FORK, forward-looking Actor, as an add-on to Actor-Critic algorithms. The evaluation of six environments demonstrated the significant performance improvements by adding FORK to two state-of-the-art model-free reinforcement learning algorithms. A variation of TD3-FORK further solved BipedalWalkerHardcore in as few as four hours with a single GPU.

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

## A ADDITIONAL DETAILS OF FORK

### A.1 REVISED REWARD NETWORK

We found from our experiments that the reward network can more accurately predict reward $r_t$ when including the next state $s_{t+1}$ as input into the reward network. Figure 5 shows the mean-square-errors (MSE) of the reward network with $(s_t, a_t)$ as the input versus with $(s_t, a_t, s_{t+1})$ as the input for BipedalWalker-v3 during the first 10,000 steps. We can clearly see that MSE is lower in the revised reward network.

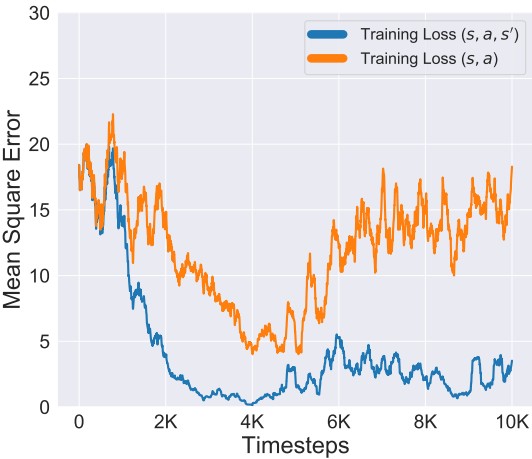

Figure 5: Training losses under the two different reward networks

### A.2 ADAPTIVE WEIGHTS VERSUS FIXED WEIGHTS

We compared TD3-FROK with the fixed weights, named as TD3-FORK-F, where the weight is chosen to be 0.4. TD3-FORK performs the best in four out of the six environments. TD3-FORK-F has a worse performance than TD3 on Walker2d-v3. We therefore proposed and used the adaptive weight because of this observation.

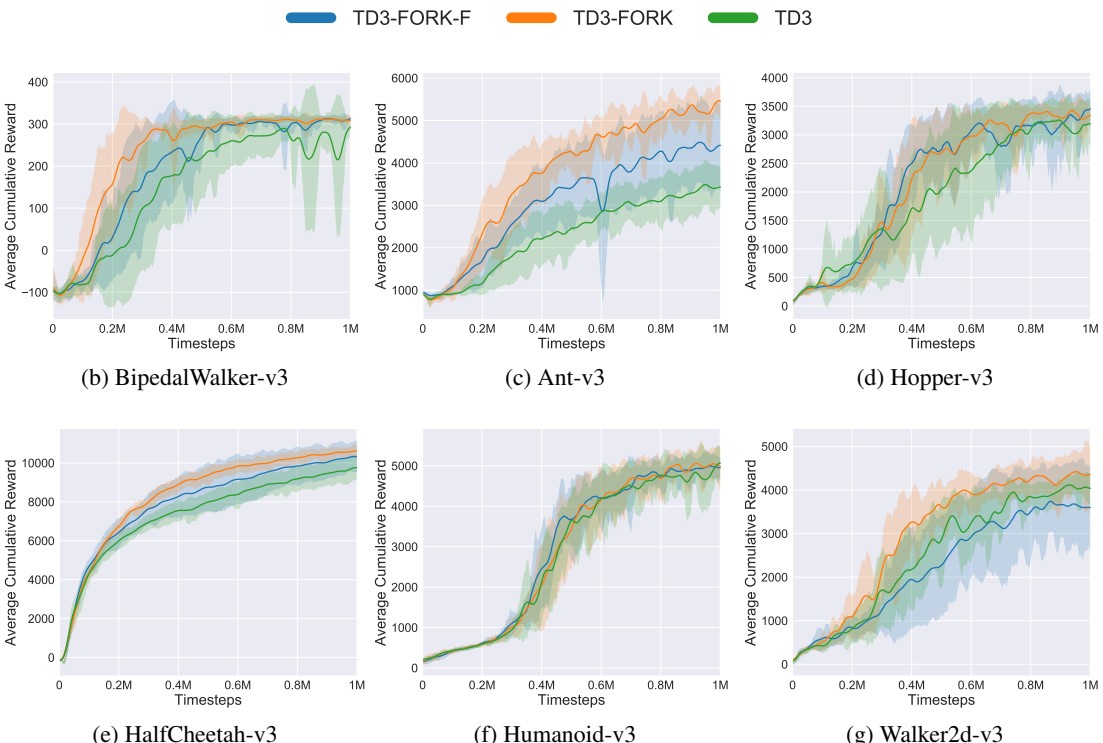

Figure 6: Learning curves of TD3, TD3-FORK-F and TD3-FORK. Curves are smoothed uniformly for visual clarity.

## A.3   TD3-FORK

The detailed description of TD3-FORK can be found in Algorithm 1, and the codes are also submitted as a supplemental material.

## A.4   HYPERPARAMETERS

Table 3 lists the hyper-parameter used in DDPG, SAC, SAC-FORK and TD3-FORK. We kept the same hyperparamter values used in SAC and TD3 codes provided or recommended by the authors. We did not tune these parameters because the goal is to show that FORK is a simple yet powerful add-on to existing Actor-Critic algorithms.

Table 4 summarizes the environment specific parameters. In particular, the base weight and base cumulative reward used in implementing the adaptive weight, and threshold for adding FORK. The base cumulative rewards for TD3-FORk are the typical cumulative rewards under TD3 after training Actor for 0.5 million steps. The base cumulative rewards for SAC-FORK are similarly chosen but with a more careful tuning. The thresholds are the typical loss values after training the system networks for about 20,000 times including the first 10,000 exploration steps. In the implementation, FORK is added to Actor training only after the system network can predict the next state reasonably well. We observed TD3-FORK with our intuitive choices of

---

**Algorithm 1** TD3-FORK

---

    Initialize critic network $Q_{\psi_1}, Q_{\psi_2}$ system network $F_\theta, R_\eta$ and actor network $A_\phi$ with random parameters $\psi_1, \psi_2, \theta, \eta, \phi$

1: Initialize target network $\phi' \leftarrow \phi, \psi_1' \leftarrow \psi_1, \psi_2' \leftarrow \psi_2$

    Initialize replay buffer $\mathcal{B}$, soft update parameter $\tau$

    Initialize base reward $r_0, w_0$, threshold $\bar{l}$ and moving average reward $\bar{r} \leftarrow 0$

    Initialize noise clip bound $c$, state bound $(o_{\min}, o_{\max})$

2: **for** episode $e = 1, \ldots, M$ **do**

3:     Initialize observation state $s_0$

4:     Initialize episode reward $r = 0$

5:     **for** $t = 1, \ldots, T$ **do**

6:         Select action $a_t$ according to the current policy and exploration noise $a_t \sim A_\phi(s) + \epsilon_t$, where $\epsilon_t \sim \mathcal{N}(0, \sigma)$

7:         Execute action $a_t$ and observe reward $r_t$, new state $s_{t+1}$

8:         Store transition tuple $(s_t, a_t, r_t, s_{t+1})$ into replay buffer $\mathcal{B}$

9:         Sample a random minibatch of $N$ transitions $(s_i, a_i, r_i, s_{i+1})$ from $\mathcal{B}$

10:        $\tilde{a}_i \leftarrow \pi_{\phi'}(s_{i+1}) + \epsilon, \epsilon \sim clip(\mathcal{N}(0, \tilde{\sigma}), -c, c))$

11:        Set $y_i = r_i + \gamma \min_{j=1,2} Q_{\psi_i'}(s_{i+1})$

12:        $r \leftarrow r + r_t$

13:        Update critic network by minimizing the loss: $L(\psi) = \frac{1}{N} \sum_{j=1}^{2} \sum_i \left(y_i - Q_{\psi_j}(s_i, a_i)\right)^2$

14:        Update state system network by minimizing loss: $L(\theta) = \|s_{i+1} - F_\theta(s_i, a_i)\|_{\text{smooth L1}}$

15:        Update reward system network by minimizing the loss: $L(\eta) = \frac{1}{N} \sum_i \left(r_i - R_\eta(s_i, a_i, s_{i+1})\right)^2$

16:        **if** $t \bmod d$ **then**

17:           Update $\phi$ by the sampled policy gradient:

18:           **if** $L(\theta) > \bar{l}$ **then**

19:             $\nabla_\phi L(\phi) = \frac{1}{N} \sum_i \nabla_a Q_{\psi_1}(s_i, a)|_{a=A_\phi(s_i)} \nabla_\phi A_\phi(s_i)$

20:           **else**

21:             $s_{i+1}' = clip(F_\theta(s_i, A_\phi(s_i)), o_{\min}, o_{\max}), s_{i+2}' = clip(F_\theta(s_{i+1}', A_\phi(s_{i+1}')), o_{\min}, o_{\max})$

22:             $\nabla_\phi L(\phi) = \frac{1}{N} \sum_i \big(\nabla_a Q_{\psi_1}(s_i, a)|_{a=A_\phi(s_i)} \nabla_\phi A_\phi(s_i) + w\nabla_a R_\eta(s_i, a, s_{i+1}')|_{a=A_\phi(s_i)} \nabla_\phi A_\phi(s_i)$

23:             $+ w\gamma\nabla_a R_\eta(s_{i+1}', a, s_{i+2}')|_{a=A_\phi(s_{i+1}')} \nabla_\phi A_\phi(s_{i+1}') + w\gamma^2 \nabla_a Q_{\psi_1}(s_{i+2}', a)|_{a=A_\phi(s_{i+2}')} \nabla_\phi A_\phi(s_{i+2}')\big)$

24:           **end if**

25:           Update target networks:

26:           $\phi' \leftarrow \tau\phi + (1 - \tau)\phi'$

27:           $\psi_i' \leftarrow \tau\psi_i + (1 - \tau)\psi_i'$

28:        **end if**

29:     **end for**

30:     Update $\bar{r} \leftarrow ((e - 1)\bar{r} + r)/e$

31:     Update adaptive weight $w \leftarrow \min(1 - \max(0, \frac{\bar{r}}{r_0}), 1)w_0$

32: **end for**

---

hyperparameters worked well across different environments and required little tuning, while SAC-FORK required some careful tuning on choosing the base weights and the base cumulative rewards.

Table 3: Hyperparameters

| Parameter | Value |
|---|---|
| Shared | |
|    optimizer | Adam |
|    learning rate | $3 \times 10^{-4}$ |
|    discount $(\gamma)$ | 0.99 |
|    replay buffer size | $10^6$ |
|    number of hidden layers (all networks) | 2 |
|    Batch Size | 100 |
|    Target Update Rate | $5 \times 10^{-3}$ |
|    Target update delay (TD3, TD3-FORK) | 2 |
|    nonlinearity | ReLU |
|    number of hidden units per layer (Critic) | 256 |
|    number of hidden units per layer (Actor) | 256 |
| TD3-FORK | |
|    number of hidden units of the system network | $[400, 300]$ (Humanoid$[1024, 1024]$) |
|    number of hidden units of the reward network | $[256, 256]$ (Humanoid$[1024, 1024]$) |
| SAC-FORK | |
|    number of hidden units of the system network | $[512, 512]$ (Humanoid$[1024, 1024]$) |
|    number of hidden units of the reward network | $[512, 512]$ (Humanoid$[1024, 1024]$) |

Table 4: Environment Specific Parameters

| Environment | Base Weight $w_0$ | Base Cumulative Reward $r_0$ | System threshold $l$ |
|---|---|---|---|
| TD3-FORK | | | |
|    BipedalWalker-v3 | 0.6 | 320 | 0.01 |
|    Ant-v3 | 0.6 | 6200 | 0.15 |
|    Hopper-v3 | 0.6 | 3800 | 0.0020 |
|    HalfCheetah-v3 | 0.6 | 12000 | 0.20 |
|    Humanoid-v3 | 0.6 | 5200 | 0.20 |
|    Walker2d-v3 | 0.6 | 4500 | 0.15 |
| SAC-FORK | | | |
|    BipedalWalker-v3 | 0.40 | 320 | 0.01 |
|    Ant-v3 | 0.40 | 5200 | 0.020 |
|    Hopper-v3 | 0.10 | 2500 | 0.0020 |
|    HalfCheetah-v3 | 0.10 | 6000 | 0.10 |
|    Humanoid-v3 | 0.10 | 4500 | 0.10 |
|    Walker2d-v3 | 0.10 | 3000 | 0.15 |

# B ADDITIONAL EXPERIMENTAL RESULTS

## B.1 BEST AVERAGE CUMULATIVE REWARD, STANDARD-DEVIATION, AND BEST INSTANCE CUMULATIVE REWARD

Table 5 summarizes the best average cumulative rewards, the associated standard-deviations, and best instance cumulative rewards. They are defined as follows. Recall that each algorithm is trained for five instances, where each instance includes 0.5 million steps of Actor training. During the training process, we evaluated the algorithm every 5,000 steps without the exploration noise. For each evaluation, we calculated

the average cumulative rewards (without discount) over 10 episodes, where each episode is $0 \sim 1,600$ under BipedalWalker-v3, is $0 \sim 1,000$ under Ant-v3, Walker2d-v3, Hopper-v3, Humanoid-v3, and is exactly 1,000 under HalfCheetah-v3.

Now let $X_\tau^{(l)}$ denote the average cumulative reward at the $\tau$th evaluation during the $l$th instance. Then

$$\text{Best Average Cumulative Reward (Best Average): } \max_\tau \frac{1}{5} \sum_{l=1}^{5} X_\tau^{(l)}$$

$$\text{Standard-Deviation: } \sqrt{\frac{1}{5} \sum_{l=1}^{5} \left( X_\tau^{(l)} - \overline{X}_\tau \right)^2}$$

$$\text{Best Instance Cumulative Reward (Best Instance): } \max_l \max_\tau X_\tau^{(l)}$$

Table 5: Best Average Cumulative Rewards, Standard Deviations, and Best Instance Cumulative Rewards of TD3-FORK, TD3, DDPG, SAC, SAC-FORK over Six Environments. The Best Value for Each Environment is in Bold Text.

| Environment | TD3-FORK | TD3 | DDPG | SAC | SAC-FORK |
|---|---|---|---|---|---|
| BipedalWalker-v3 | | | | | |
| Best Average Cumulative Reward | **317.40** | 307.69 | 125.90 | 223.20 | 303.18 |
| Standard Deviation | ±4.68 | ±11.18 | ±130.06 | ±120.36 | ±13.58 |
| Best Instance Cumulative Reward | **322.97** | 317.47 | 254.17 | 314.91 | 322.58 |
| Ant-v3 | | | | | |
| Best Average Cumulative Reward | **5699.37** | 3652.11 | 995.59 | 1892.50 | 3091.00 |
| Standard Deviation | ±234.62 | ±510.50 | ±2.67 | ±523.40 | ±1000 |
| Best Instance Cumulative Reward | **6015.47** | 4546.47 | 999.79 | 2595.72 | 4134.06 |
| Hopper-v3 | | | | | |
| Best Average Cumulative Reward | **3522.57** | 3517.57 | 1713.52 | 3381.01 | 3359.18 |
| Standard Deviation | ±120.22 | ±72.37 | ±957.20 | ±164.42 | ±119.30 |
| Best Instance Cumulative Reward | **3659.27** | 3591.42 | 3573.53 | 3595.08 | 3511.14 |
| HalfCheetah-v3 | | | | | |
| Best Average Cumulative Reward | **10818.89** | 9893.04 | 5886.75 | 9314.64 | 8781.06 |
| Standard Deviation | ±174.77 | ±679.21 | ±2499.82 | ±598.47 | ±816.97 |
| Best Instance Cumulative Reward | **11044.74** | 10361.92 | 9595.08 | 9802.70 | 9848.15 |
| Humanoid-v3 | | | | | |
| Best Average Cumulative Reward | **5439.31** | 5386.21 | 456.35 | 5293.89 | 5233.89 |
| Standard Deviation | ±152.26 | ±115.92 | ±75.22 | ±26.75 | ±65.38 |
| Best Instance Cumulative Reward | **5685.77** | 5513.46 | 529.84 | 5333.69 | 5284.72 |
| Walker2d-v3 | | | | | |
| Best Average Cumulative Reward | **4616.46** | 4278.90 | 2404.60 | 4037.87 | 4243.69 |
| Standard Deviation | ±499.26 | ±195.35 | ±1359.97 | ±740.40 | ±467.76 |
| Best Instance Cumulative Reward | **5192.52** | 4541.51 | 4618.44 | 5116.54 | 5042.80 |

## B.2 COMPARISON WITH THE SAME AMOUNT OF TOTAL TRAINING

In Section 4, the algorithms were compared assuming the same amount of Actor training since our focus is on the performance of Actor. Since TD3 uses delayed Actor training, Critic of TD3 and TD3-FORK is trained twice as much as Critic of SAC and SAC-FORk when Actor is trained the same number of steps, which gives advantage to TD3 and TD3-FORK.

To further compare the performance of TD3-FORK and SAC-FORK, we present the results where for each algorithm, Actor and Critic, together, were trained 1.5 million steps. In particular, Actor was trained 0.5 million steps and Critic is trained 1 million steps under TD3 and TD3-FORK; and Actor and Critic were trained 0.75 million steps each under SAC and SAC-FORK. The results can be found in Figure 7.

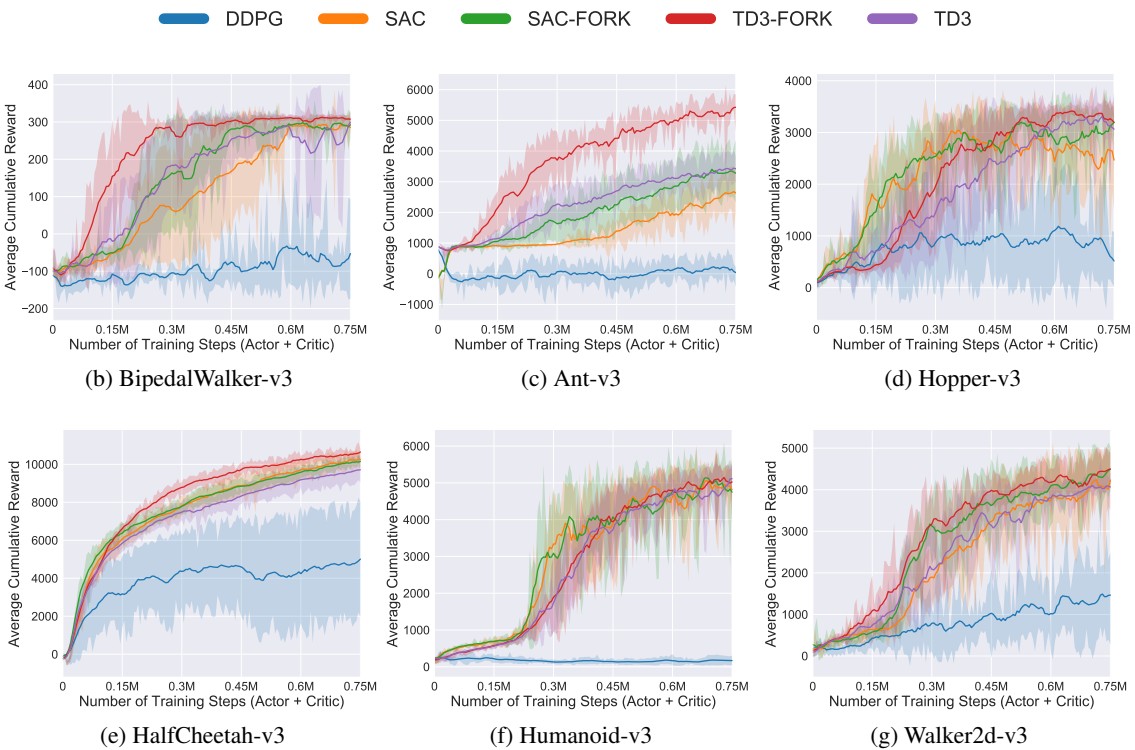

Figure 7: Learning curves of the six environments. Under each algorithm, Actor and Critic, together, were trained for 1.5 million steps. Curves are smoothed uniformly for visual clarity.

Table 6 summarizes the best average cumulative rewards, standard-deviations, and the best instance cumulative rewards. We can see that in terms of the best average cumulative rewards, TD3-FORK performed the best in four out of the six environments, including BipedalWalker, Ant, Hopper and HalfCheetah; and SAC-FORK performed the best in the remaining two — Humanoid and Walker2d.

## B.3 DIFFERENT IMPLEMENTATIONS OF FORK

As we mentioned in Section 3, FORK can have different implementations. We considered two examples in Section 4, and compared their performance as add-on to TD3. We call FORK with loss function Equation (5)

Table 6: Best Average Cumulative Reward, Standard-Deviation, and Best Instance Cumulative Reward. The Best Value for Each Environment is in Bold Text.

| Environment | TD3-FORK | TD3 | DDPG | SAC | SAC-FORK |
|---|---|---|---|---|---|
| BipedalWalker-v3 | | | | | |
| Best Average Cumulative Reward | **317.40** | 307.69 | 125.90 | 304.56 | 311.49 |
| Standard Deviation | ±4.68 | ±11.18 | ±130.06 | ±13.18 | ±9.26 |
| Best Instance Cumulative Reward | 322.97 | 317.47 | 254.17 | 319.16 | **325.56** |
| Ant-v3 | | | | | |
| Best Average Cumulative Reward | **5699.37** | 3652.11 | 995.59 | 3108.41 | 3897.19 |
| Standard Deviation | ±234.62 | ±510.50 | ±2.67 | ±663.05 | ±905.42 |
| Best Instance Cumulative Reward | **6015.47** | 4546.47 | 999.79 | 4048.27 | 5080.86 |
| Hopper-v3 | | | | | |
| Best Average Cumulative Reward | **3522.57** | 3517.57 | 1713.52 | 3381.01 | 3407.15 |
| Standard Deviation | ±120.22 | ±72.37 | ±957.20 | ±164.42 | ±138.286 |
| Best Instance Cumulative Reward | **3659.27** | 3591.42 | 3573.53 | 3595.08 | 3592.56 |
| HalfCheetah-v3 | | | | | |
| Best Average Cumulative Reward | **10818.89** | 9893.04 | 5886.75 | 10440.85 | 9704.72 |
| Standard Deviation | ±174.77 | ±679.21 | ±2499.82 | ±415.20 | ±1213.09 |
| Best Instance Cumulative Reward | **11044.74** | 10361.92 | 9595.08 | 10737.45 | 10505.13 |
| Humanoid-v3 | | | | | |
| Best Average Cumulative Reward | 5439.31 | 5386.21 | 456.35 | 5415.61 | **5443.04** |
| Standard Deviation | ±152.26 | ±115.92 | ±75.22 | ±53.38 | ±99.44 |
| Best Instance Cumulative Reward | **5685.77** | 5513.46 | 529.84 | 5490.09 | 5575.66 |
| Walker2d-v3 | | | | | |
| Best Average Cumulative Reward | 4616.46 | 4278.90 | 2404.60 | 4468.58 | **4693.66** |
| Standard Deviation | ±499.26 | ±195.35 | ±1359.97 | ±622.17 | ±468.90 |
| Best Instance Cumulative Reward | 5192.52 | 4541.51 | 4618.44 | 5466.40 | **5514.77** |

FORK-S, standing for single-step FORK; call FORK with loss function Equation (6) and $w' = 0.5$ FORK-DQ, standing for Double-Q FORK; and FORK with loss function Equation (6) and $w' = 0$ FORK-Q, standing for Q FORK. From Table 7, we can see that in terms of best average cumulative reward, TD3-FORK performs the best four out of the six environments and TD3-FORK-S performs the best in the remaining two. This is the reason we selected the current form of FORK.

## C BIPEDALWALKERHARDCORE

TD3-FORK-DQ can solve the difficult BipedalWalker-Hardcore-v3 environment with as few as four hours. The hardcore version is much more difficult than BipedalWalker. For example, a known algorithm needs to train for days on a 72 cpu AWS EC2 with 64 worker processes taking the raw frames as input (`https://github.com/dgriff777/a3c_continuous`). TD3-FORK-DQ, a variation of TD3-FORK, can solve the problem in as few as four hours by using default GPU setting provided by Google Colab[2] and with sensory data (not images). The performance on BipedalWalkerHardcore-v3 during and after training can be viewed at `https://youtu.be/0nYQpXtxh-Q`. The codes have been submitted as a supplementary materials.

To solve BipedalWalkerHardcore, we made several additional changes.

---

[2]https://colab.research.google.com/notebooks/intro.ipynb

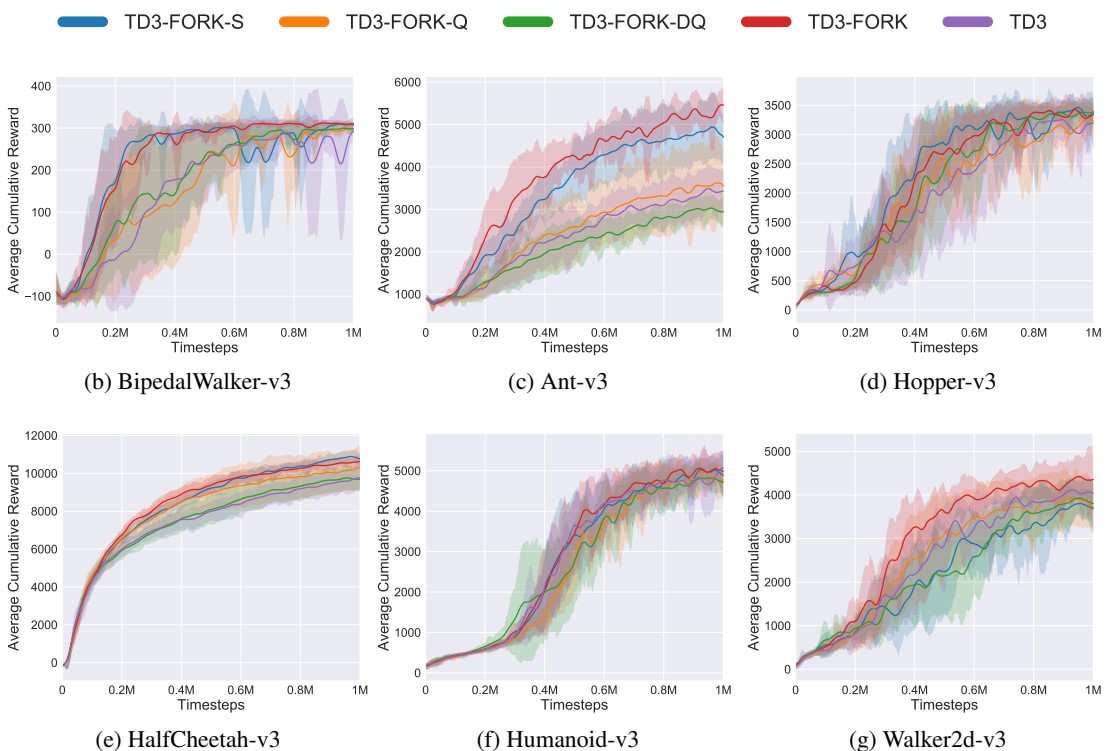

Figure 8: Learning curves of TD3-FORK, TD3, TD3-FORK-S, TD3-FORK-Q and TD3-FORK-DQ. Curves are smoothed uniformly for visual clarity.

(i) We changed the -100 reward to -5.

(ii) We increased other rewards by a factor of 5.

(iii) We implemented a replay buffer where failed episodes, in which the bipedalwalker fell down at the end, and successful episodes are added to the replay-buffer with 5:1 ratio.

The changes to the rewards (i) and (ii) were suggested in the blog [3]. Using reward scaling to improve performance has been also reported in (Henderson et al., 2017). We made change (iii) because we found failed episodes are more useful for learning than successful ones. The reason we believe is that when the bipidedalwalker already knows how to handle a terrain, there is no need to further train using the same type of terrain. When the training is near the end, most of the episodes are successful so adding these successful episodes overwhelm the more useful episodes (failed ones), which slows down the learning.

---

[3]https://mp.weixin.qq.com/s?__biz=MzA5MDMwMTIyNQ==&mid=2649294554&idx=1&sn=9f893801b8917575779430cae89829fb&scene=21#wechat_redirect

Table 7: Best Average Cumulative Rewards, Standard Deviations, and Best Instance Cumulative Rewards of TD3-FORK, TD3, TD3-FORK-S, TD3-FORK-Q, and TD3-FORK-DQ over Six Environments. The Best Values are in Bold Text.

| Environment | TD3-FORK | TD3 | TD3-FORK-S | TD3-FORK-Q | TD3-FORK-DQ |
|---|---|---|---|---|---|
| BipedalWalker-v3 | | | | | |
| Best Average | **317.40** | 307.69 | 314.63 | 302.96 | 306.44 |
| Standard Deviation | ±4.68 | ±11.18 | ±3.61 | ±9.17 | ±5.98 |
| Best Instance | **322.97** | 317.47 | 320.11 | 313.93 | 315.55 |
| Ant-v3 | | | | | |
| Best Average | **5699.37** | 3652.11 | 5226.37 | 3905.39 | 3287.929 |
| Standard Deviation | ±234.62 | ±510.50 | ±748.03 | ±1019.78 | ±220.10 |
| Best Instance | **6015.47** | 4546.47 | 5748.94 | 5563.98 | 3499.86 |
| Hopper-v3 | | | | | |
| Best Average | 3522.57 | 3517.57 | **3605.09** | 3426.54 | 3482.14 |
| Standard Deviation | ±120.22 | ±72.37 | ±89.04 | ±119.51 | ±136.36 |
| Best Instance | 3659.27 | 3591.42 | **3740.71** | 3588.89 | 3677.66 |
| HalfCheetah-v3 | | | | | |
| Best Average | 10818.89 | 9893.04 | **11077.10** | 10405.40 | 9942.23 |
| Standard Deviation | ±174.77 | ±679.21 | ±337.05 | ±1154.00 | ±675.52 |
| Best Instance | 11044.74 | 10361.92 | 11450.83 | **11524.62** | 10668.92 |
| Humanoid-v3 | | | | | |
| Best Average | **5439.31** | 5386.21 | 5345.92 | 5255.54 | 5270.73 |
| Standard Deviation | ±152.26 | ±115.92 | ±204.92 | ±210.65 | ±96.97 |
| Best Instance | 5685.77 | 5513.46 | **5706.35** | 5669.17 | 5499.36 |
| Walker2d-v3 | | | | | |
| Best Average | **4616.46** | 4278.90 | 4089.02 | 4175.40 | 4177.34 |
| Standard Deviation | ±499.26 | ±195.35 | ±260.61 | ±601.82 | ±372.90 |
| Best Instance | **5192.52** | 4541.51 | 4450.75 | 4934.03 | 4920.74 |

