# OpenReview forum: "FORK: A FORward-looKing Actor for Model-Free Reinforcement Learning"
_ICLR.cc/2021/Conference — Reject_

### Official Review · AnonReviewer2 · 2020-10-25

**Rating:** 5
**Confidence:** 4

**Review:**

The paper proposes to combine ideas from model-based RL into model-free off-policy policy gradient algorithms (like SAC and TD3). Specifically, the paper proposes to learn auxiliary models of environment rewards and dynamics and use a two-step rollout from these models during the computation of the policy gradient. The paper presents results of this mechanism applied to standard SAC and TD3 implementations on a variety of continuous control environments with favorable results.

Strengths:

-- The story is generally easy to follow.

-- I appreciate that the authors didn't just evaluate on the extremely common (and somewhat saturated) MuJoCo benchmarks, and presented additional results on Bipedal Walker.

-- As far as I can tell, the policy update proposed here is novel.

Weaknesses:

-- While the policy update appears novel, it is very similar to techniques in the model-based RL literature. Given this, it would greatly improve the paper if it had appropriate comparisons to similar model-based techniques, especially those which claim to combine model-free updates with model-based techniques; for example https://arxiv.org/abs/1906.08253

-- Moreover, experimentally it would be nice to see comparisons to state-of-the-art model based RL methods. For example, https://arxiv.org/abs/1802.10592

-- In terms of the current experiment results, I found the conclusions favorable to the proposed technique, but not a very compelling demonstration. For example, in Table 1, almost all the environments produce only a *slight* benefit for the proposed method. It appears the only significant benefit is on Ant. Similarly in Table 2, we see the results of SAC-FORK are sometimes much worse than SAC on its own.

-- In terms of motivation for the method, I was not entirely convinced of why the proposed update is needed. The paper appeals to the idea of needing to reason about values in the future. But shouldn't the Q-value already encapsulate this? Moreover, the proposed update ends up only optimizing *actions* in the future, rather than somehow reasoning about the values at steps t+1, t+2 to decide the best action at step t.

---

### Official Review · AnonReviewer1 · 2020-10-27
**Incorrect algorithm and insufficient discussion of other model based RL algorithms**

**Rating:** 3
**Confidence:** 5

**Review:**

**Correctness Issue**: The loss in equation 2 is the sum of the regular actor loss using the critic + an n-step return version of policy value, which would be reasonable for a policy update. However, the expression provided for the gradient (and the calculations in the code) are incorrect in that they are not the gradient of equation 2 with respect to the policy. In particular, it is not reflective of policy performance, as it fails to account for the fact that the future states in the imagined rollout are also functions of the policy.
The resulting policy update in FORK actually consists of:
1. Regular actor update using the critic at the current state
2. Updating to greedily maximize the reward at each intermediate timestep (which does not reflect the policy performance)
3. Another regular actor update from the last state in the rollout.

The straightforward way of correcting this to make it optimize the loss would be to simply differentiate through the model, which may not work well as differentiating through learned models often leads to instability (though it may be fine for such short rollouts). Alternatively, a REINFORCE estimator can instead be used for the gradient of the n-step term for stochastic policies.

We note that the loss in eq. 6 (if we keep the dependence of the future state on the policy) is also not reflective of policy returns (since it overweights future returns over the present), but can provide reasonable interpretation of the FORK update if we drop the dependence of the future state on the policy as the authors do.
The gradient of the FORK-Q variant the authors take from eq.6 is the actually the update we would obtain by using the regular actor update (the $\nabla_{\theta} \pi_{\theta}(s) Q(s, \pi(s))$ term for deterministic policies) but under a different state distribution. Instead of using the state distribution in the replay buffer, the update takes into account the distribution obtained by running the policy for a few steps starting from the buffer distribution. As running the policy from the buffer distribution would result in a distribution closer to the on-policy state distribution, one explanation for why the FORK-Q update would improve is that the gradient is closer to the on-policy policy gradients by changing the state distribution.

The same reasoning can be applied to primary FORK update presented, as it includes an actor update on the value in future states (3rd term in my previous list). The issue is with the greedy reward maximization in the inner steps (2nd term), but maybe it simply doesn't hurt on the environments tested or it perhaps takes advantage of a bias-variance tradeoff in greedily maximizing immediate reward for a few intermediate timesteps. I would like the authors to explicitly address these issues in the paper and present a clear explanation of why their FORK should be a better policy update.

**Relation with Model Based RL Methods**: Overall, I also disagree with the authors' claims that FORK is very different and much simpler than other model based algorithms. With regards to their comment on how their method is somehow simpler? than rollout based methods, their policy update is using a short Monte-Carlo rollout for estimating the policy gradients; the difference being that the rollouts are being used to update a policy rather than to explicitly plan at test time.

The authors' claim that FORK does not require high-fidelity simulation seems unsupported to me, and the claim that model-based RL algorithms use the model in a sophisticated way is vague. In particular, Dyna-style algorithms (like STEVE https://arxiv.org/abs/1807.01675, MBPO https://arxiv.org/abs/1906.08253) which use the model to generate experience to help learn the critic, seem to be using the model in the same way as FORK (in the sense that they only use the model to generate samples with short rollouts). There is also no discussion of methods like ME-TRPO https://arxiv.org/abs/1802.10592, SLBO https://arxiv.org/abs/1807.03858, or the algorithms in https://arxiv.org/abs/2004.07804, all of which only the model to generate trajectories to use with a policy gradient algorithm. Overall, I would appreciate much more discussion about how FORK relates to and compares empirically against past RL algorithms that only use the model to generate samples, as well as clarifying the statements about FORK uses the model in a less sophisticated way.

On a separate note, using the model to generate n-step-return estimates of the policy value has been previously done in https://arxiv.org/abs/1807.01675 for example. The key difference is that prior work used it to generate target values for learning the Q-function, while here it is used purely for policy updates. Given how similarly the models are used however, I would recommend discussing this line of work explicitly in the related work, even though they are complementary.

**Experimental Evaluation**: Despite the aforementioned correctness issue, the method seems to provide improvements when applied to TD3, and seemingly smaller improvements on top of SAC. However, I find it extremely strange that the authors chose in Figure 4 to plot returns against the number of training steps instead of the number of samples. As acknowledged by the authors, this makes the SAC vs TD3 comparisons incomparable, with TD3 and TD3-FORk enjoying the advantage of having seen twice as many samples. Moreover, comparing only on the number of actor/critic updates isn't even a fair comparison between FORK and baselines, as FORK additionally has to train a dynamics model and reward predictor. I would highly recommend simply showing learning curves with respect to the actual number of environment samples, rather than arbitrarily using the number of actor updates.

I would also like to see comparisons against model based RL baselines, particularly MBPO, which uses a Dyna-style update with SAC to compare which method of utilizing the model is better. The authors could also see if the FORK actor update further improves upon MBPO or other model based RL methods.

**Summary**:  As it stands, I believe the paper should be rejected due to the correctness issues (and resulting lack of justification for why FORK should give better actor updates), and insufficient discussion of how it relates to prior in model based RL. To consider accepting the paper, I would at least need to see these issues addressed by the authors.

Regarding novelty and significance, using a model to predict n-step value estimates (as the paper claims to be doing) or using the model to explicitly adjust the state distribution of the policy update (as I suspect this might be doing instead) for the policy update in an off-policy actor critic algorithm has not been done before as far as I know. However, this change in how the model is used seems fairly minor, and to be convinced it were useful, I would like to see evidence of how it compares against the other model based RL algorithms. In particular, the benefit I imagine it might have over other model based RL methods is in being more robust to poorly fit models, but I would need to see empirical evidence supporting this.

---

### Official Review · AnonReviewer4 · 2020-10-29
**The author uses a model based approach to improve the bias in Q network training**

**Rating:** 5
**Confidence:** 4

**Review:**

Summary:

The authors proposed a simple modification to popular off policy algorithms, such as SAC and TD3. By employing a model network and reward network, the authors can expand the bellman update operation using predicted next few steps, similar to the GAE (general advantage estimation). The authors demonstrated that algorithms such as TD3, DDPG, and SAC can all benefit from their approach.


Pros:
The paper is clearly written and easy to understand.

The concept is simple and the implementation is straight forward. The results indicated that the training sample efficiency and final policy performance of the tested algorithms have improved for 6 benchmark tasks, compared with their "vanilla" version baseline.


Cons:
While it is generally understandable that GAE-like approaches can help balance between bias and variance in the Q-value estimation, I am not convinced if the proposed networks are needed. The authors used two additional networks: the model network (they call "system network") and the reward network. The model network computes the standard transition s_t, a_t -> s_t+1, and the reward network estimates the true reward r = R(s_t, a_t, s_t+1). However, as reward function is generally provided, I don't see an additional reward network is necessary here. Second, in off-policy learning, the state transition and next states are already known, instead of using a model network to predict the next states, one can simply sample a small trajectory containing multiple consecutive state-action transitions, and use them to do the Q-value estimation.

Recommendations:
To address the concern above, I propose the authors to add two more ablation studies: (1) Remove the reward network and only use the reward function. (2) Remove both the reward network and the model/system network, and use the recorded future states to estimate the Q-values.

---

### Official Review · AnonReviewer3 · 2020-10-29
**Model-based RL on top of actor-critic method**

**Rating:** 3
**Confidence:** 5

**Review:**

### Summary
This paper focuses on the field of off-policy reinforcement learning. Specifically, the authors propose a model-based reinforcement learning method on top of actor-critic methods. The proposed method trains a dynamics model and a reward function on the off-policy data with supervised learning, and then uses the trained model to generate synthetic future states and rewards during the actor update. During policy update for a given state, the method computes the sum of the Q value estimate of the state and the Q value expansion for a few steps using the learned dynamics model and reward function.

The authors implement the proposed method on top of SAC, and TD3, and evaluate their performances on several MuJoCo and Box2D environments. The experiment results show that the proposed method outperforms the model-free baseline in terms of sample efficiency.

### Comments
The paper is well written and the idea proposed in this paper is really easy to understand. The authors also include a wide suite of experiments to demonstrate the sample efficiency of the proposed method. Despite these advantages, I cannot recommend acceptance of this paper due to the lack of novelty and absence of fair baseline comparison, which I will elaborate on next.

First of all, despite the title of the paper, the proposed method is really a model-based reinforcement learning method since the system network and reward network are just dynamics model and reward model. The proposed objective for the policy (eqn 2), is merely a sum of current Q value estimate and Q value expansion for a few steps using the learned model, which has been proposed before in various papers such as [1] and [2]. The only difference is that when computing the gradient with respect to the policy, the authors leave out the gradient that passes through the learned model, which results in biased estimate of the policy gradient. Therefore, I’m not convinced about the novelty of the proposed method.

Moreover, while proposing a model-based method. The authors do not include baseline comparisons with other model-based RL methods. It is widely known that on low-dimensional control tasks, model-based method outperforms model-free methods ([3]), and therefore merely comparing to model-free baselines is unfair. It would be important to include comparisons to model based methods ([3]).

Due to the lack of novelty and fair comparison to existing model-based methods, I cannot recommend acceptance for this paper.



References

[1] Heess, Nicolas, et al. "Learning continuous control policies by stochastic value gradients." Advances in Neural Information Processing Systems. 2015.

[2] Clavera, Ignasi, Yao Fu, and Pieter Abbeel. "Model-Augmented Actor-Critic: Backpropagating through Paths." International Conference on Learning Representations. 2019.

[3] Langlois, Eric, et al. "Benchmarking model-based reinforcement learning." arXiv preprint arXiv:1907.02057 (2019).

---

### Decision · Program_Chairs · 2021-01-07
**Final Decision**

**Decision:**

Reject

**Comment:**

All reviewers agreed that the novelty of the method was not at the level expected for publication, and also raised a number of technical concerns regarding the approach. There was no response from the authors on these issues, hence the reviewer consensus is that the paper is not ready for publication at this time.